# Multiple Pregnancy and the Risk of Postpartum Hemorrhage: Retrospective Analysis in a Tertiary Level Center of Care

**DOI:** 10.3390/diagnostics13030446

**Published:** 2023-01-26

**Authors:** Giulia di Marco, Elisa Bevilacqua, Elvira Passananti, Caterina Neri, Chiara Airoldi, Alessia Maccarrone, Vittoria Ciavarro, Antonio Lanzone, Alessandra Familiari

**Affiliations:** 1Department of Woman and Child Health and Public Health, Fondazione Policlinico Universitario A. Gemelli IRCCS, 00168 Rome, Italy; 2Department of Translational Medicine, University of Eastern Piedmont, 13100 Novara, Italy

**Keywords:** multiple pregnancy, twins, postpartum hemorrhage

## Abstract

The aim of our study was to identify characteristics associated with postpartum hemorrhage (PPH defined as blood loss >1000 mL) in twin pregnancies in order to select patients at higher risk to be treated. This retrospective study includes multiple pregnancies between 2015 and 2020. The possible association between pregnancy characteristics and the primary endpoint (occurrence of PPH) was conducted using chi-square or Fisher exact test and Wilcoxon test. Then, univariate logistic models were performed considering as outcome the PPH, and the odds ratios with 95% CI were estimated. Finally, a multivariate logistic model was implemented, including all significant covariates. Seven hundred seven twin pregnancies giving birth beyond 32 weeks were included and of those, 120 (16.97%) had a PPH. The univariate analysis showed that factors significantly associated with PPH were: Preterm delivery, episiotomy, neonatal weight, and mode of delivery. The multivariate analysis showed that the most important factors were episiotomy and neonatal weight. The results show that the performance of episiotomy and the neonatal weight are the factors that most impact the risk of PPH in twin pregnancies. The correct identification of factors associated with PPH in twins could ideally allow to modify the clinical management and positively affect the rate of complications.

## 1. Introduction

Twin pregnancy represents around 1–2% [1] of all pregnancies. Its incidence has been rapidly rising in the last decades due to the increase in maternal age at first pregnancy and the constant progress of assisted reproductive technology (ART) techniques [2].

Twin pregnancies’ outcome is burdened by several maternal and fetal factors, making obstetric management until the time of delivery a real challenge. Evidence from the literature highlights that one of the most dangerous factors that need to be considered when planning a twin delivery is the risk of postpartum hemorrhage (PPH). The World Health Organization defines PPH as the loss of more than 500 mL of blood after a vaginal delivery or greater than 1000 mL after cesarean delivery [3]. It is estimated that post-partum hemorrhage is the leading cause of maternal death worldwide, resulting in up to 28% of maternal deaths [4,5] and that it is more frequent in multiple pregnancies than in singletons [6,7]. This phenomenon has been related to uterine overdistension, which can compromise myometrial contractility after delivery and increase the incidence of uterine atony, which is, in turn, the leading cause of obstetric bleeding, accounting for up to 80% of PPH cases [5,6,8]. Risk factors for uterine atony include obesity, white, or Hispanic ethnicity, polyhydramnios, preeclampsia, anemia, and chorioamnionitis, as well as multiple pregnancy [8]. Compared to singletons, in multiple gestation, maternal blood volume and uterine blood flow increase to support additional uterine, placental, and fetal tissue [9].

Women carrying twins are subject to greater maternal hemodynamic changes: Patients with multiple gestation have a 20% higher cardiac output and 10–20% greater increase in plasma volume than singletons [10]. These changes lead to an increase not only in pulmonary edema but also in other potential pregnancy complications like anemia, PPH, gestational hypertension, and preeclampsia [10,11].

Despite this, clinically significant bleeding occurs only in a subset of twin pregnancies, and the optimal way to identify those at higher risk for PPH remains uncertain.

Therefore, the aim of our study was to identify maternal and peripartum characteristics in twin pregnancies that can be eventually associated with PPH in order to select patients at higher risk. This stratification will hopefully allow the obstetricians to modify the clinical management in those high-risk twin pregnancies in order to positively affect the rate of maternal morbidity and mortality and to best counsel patients regarding the mode of delivery.

## 2. Materials and Methods

This is an observational cohort study including twin pregnancies giving birth at IRCCS Fondazione Policlinico Gemelli, Rome, between 2015 and 2020. We retrospectively collected maternal and perinatal characteristics of all twin pregnancies giving birth >32 weeks. Inclusion criteria were: All women with a twin pregnancy (bichorionic or monochorionic biamniotic with two fetuses) and birth at a gestational time of >32 weeks. We excluded all twin pregnancies whit unknown outcomes, those with age <18 years, monochorionic monoamniotic, triplet pregnancies, and those that refused to sign informed consent to study participation.

All patients admitted in labor in the delivery room were asked to give written informed consent to use the data of the pregnancy and outcome for research. All pregnancy and delivery characteristics were stored in a registry. The data extracted for the present study were anonymous, therefore, no further ethical approval was necessary. This study complies with the declaration of Helsinki. The study was approved by our local Ethical Committee (Protocol number 0002657/22).

Elective cesarean delivery has always been scheduled for fetal or maternal indications (i.e., previous cesarean delivery, breech, or other fetal presentation than cephalic for the first twin, complicated monochorionic twin pregnancy, maternal comorbidity, severe fetal growth restriction, placenta previa and/or accrete). Unscheduled cesarean delivery has been performed for all urgent indications like non-reassuring fetal heart rate, placental abruption, premature rupture of membranes with anomalous fetal presentation, cord prolapse, severe maternal hypertensive disorders, or severe fetal growth restriction.

Operative vaginal delivery was performed for several obstetric indications, with the more frequent being the delay in the delivery of the second twin (more than 20 min).

The primary outcome was the prevalence of patients with PPH, defined as estimated blood loss (EBL) >1000 mL, measured using specific graduated bags both in case of vaginal and cesarean delivery. Subsequently, we explored the possible association between PPH and maternal and gestational characteristics and/or pregnancy complications.

Descriptive statistics were conducted considering the whole population and separated for EBL (>1000 mL) (Table 1 and Table 2). Absolute and relative frequencies were reported for categorical variables, while mean and standard deviations or median and interquartile ranges were for numerical ones, as appropriate. We analyzed the variables available, and for neonatal weight, we decided to calculate the mean values among the twins, both with the max weight.

To explore the possible association between the primary outcome and the analyzed variables, comparisons between women with PPH vs. the others were conducted using chi-square or Fisher exact test for categorical variables while *t*-test or Wilcoxon test were used for numerical ones. Then, univariate logistic models were performed considering as outcome blood loss and the odds ratio (OR) with 95% confidence intervals were estimated [95% CI]. Finally, a multivariate logistic model was implemented including all significant covariates (*p* < 0.10) and excluding, step-by-step, the terms that lose their statistical significance. For the analysis, the unit of interest was the delivery/mothers.

Sensitivity analysis was conducted using different delivery categories and including the neonatal weight as the maximum or the mean value observed among twins. In the final analysis, we considered 4 categories for delivery: Scheduled cesarean delivery, unscheduled cesarean delivery, spontaneous vaginal delivery, and operative vaginal delivery, and we included only the mean weight.

All the analyses were conducted using the software SAS 9.4, and statistical significance was set to 0.05.

## 3. Results

Of the 784 consecutive twin pregnancies initially recruited, 77 were excluded for different reasons (triplets or quadruplets pregnancies, delivery <32 weeks of gestation, missing data). We finally analyzed data regarding 707 twin pregnancies giving birth beyond 32 weeks of gestation in our hospital. Of those, 120 (16.97%) had PPH. Demographic characteristics of the entire population are described in Table 1. No statistically significant differences have been reported for twin pregnancies with PPH compared to those with EBL < 1000 mL.

The univariate model shows that PPH was significantly associated with the presence of episiotomy [OR of 4.08, CI 1.76–9.46], with increasing neonatal weight [OR of 1.001, CI 1.000–1.001] and with preterm delivery (gestational age >32 and <37 weeks) [OR of 0.62, CI 0.40–0.95]. Moreover, it appears that the mode of delivery can affect the risk of PPH and in particular operative vaginal delivery shows a statistically significant association with increasing EBL [OR of 5.28, CI 1.24–22.46]. In our study population, 404 patients (57.14% of the initial recruited population) had scheduled cesarean delivery with a PPH in 72 cases (17.82%), 226 (31.97% of the 707 recruited patients) had a unscheduled cesarean delivery with 30 cases of PPH (13.7% of the all unscheduled), 68 (9.62%) had a spontaneous vaginal delivery with 13 cases (19.2%), and 9 of them had an operative vaginal delivery (1.27%) with 5 cases of PPH (55.56%), as represented in Figure 1.

It follows that vaginal and scheduled cesarean deliveries have an overlapping risk, while operative vaginal deliveries have a significantly increased risk of PPH (Table 2).

After applying the multivariate model, it appears that the two statistically significant variables are mainly the episiotomy [OR of 3.59; *p*-value 0.0037] and the mean neonatal weight [OR of 1.001; *p*-value 0.0051). In particular within the perineal tear, we have distinguished three categories: Episiotomy, spontaneous vaginal tear, and no tear, where only the first one resulted statistically significant (Table 3). In our cohort, we did not register any major maternal complications, such as thromboembolism or maternal death, so it was not possible to compare this outcome.

## 4. Discussion

Several investigations have attempted to develop predictive models for PPH, but they have been, at their best, only moderately successful, and have focused predominantly on singleton gestations [12,13]. Moreover, the prognostic value of various maternal and pregnancy characteristics in twins, including those identified as risk factors in singleton gestations, is still under debate. The clinical value of identifying twin pregnancies at increased risk of PPH relies on the possibility of creating strategies of prevention, such as the use of specific medications during labor (i.e., carbetocyne instead of oxytocine) [14,15]. Moreover, the correct stratification of the risk could potentially help in deciding the safest mode of delivery, obstetric management, and the appropriate counseling for the patient.

The findings of our study, evaluating the possible association between maternal and peripartum characteristics and the occurrence of PPH in twin pregnancies, demonstrate that among all the analyzed variables, the episiotomy, the rising mean neonatal weight, and the operative vaginal delivery are significantly associated with an increased risk of PPH, while this risk decreases when there is a preterm delivery (>32 and <37 gestational weeks).

The currently available literature investigating the role of the mode of delivery in twins on the specific outcome EBL is still unclear. There is evidence showing that cesarean delivery, especially when occurring after a trial of labor, is associated with an increased risk for PPH compared to spontaneous vaginal delivery [16,17,18]. This is intuitive if we think that cesarean delivery is abdominal surgery and carries itself an increased the risk of blood loss. On the other hand, Easter et al. report that women with twin pregnancies undergoing a trial of labor have a higher risk of maternal morbidity than those undergoing an elective cesarean delivery, primarily as a result of hemorrhage [19].

Our results show that patients with twin pregnancies undergoing cesarean delivery have a reduction in the risk of PPH. In our cohort, indeed, the cesarean delivery, whether scheduled or not, has shown a practically overlapping risk of spontaneous vaginal delivery, while the risk was particularly higher in the case of operative vaginal delivery.

Therefore, the results of our study show that the risk of PPH is strongly related to the mode of delivery as we describe 19% of PPH in twin pregnancies undergoing vaginal delivery and 18% in those cases of scheduled cesarean delivery with 14% in cases with urgent/unscheduled cesarean section. Operative vaginal delivery was associated with 56% of PPH, presenting the highest risk of PPH in our cohort of twins.

Several factors could possibly have contributed to these results. The first explanation is that the risk of PPH in twins basically depends on mechanical reasons, namely uterine overdistension. This would explain both the influence of gestational age at delivery (the earlier, the less the EBL) and of the increasing neonatal weight on the risk of PPH. Moreover, we are a referral center, and our population is burdened by a higher rate of high-risk twin pregnancies compared to the uncomplicated, low-risk twin pregnancies.

Another interesting result of our series is that twins undergoing operative vaginal delivery have a four-fold increased risk of having a PPH, mainly related to the indication for the operative delivery and to the occurrence of the episiotomy that determines apparently a poorly controlled blood loss and, in turn, to the performance of episiotomy. A reliable explanation for this result could be that a common indication for the application of vacuum is the delay in the progression of the fetal descent, often related to the increased neonatal weight, which, in turn, has been demonstrated to be an independent risk factor for PPH in twins [19,20].

Moreover, our study brought another unexpected result: Preterm birth, which usually involves multiple neonatal complications, is associated with a lower risk of PPH for the mother. Normally preterm delivery occurs as a result of maternal-fetal complications that often require an unscheduled cesarean delivery, so it comes naturally to think that the presence of an unprepared uterus with an undeveloped inferior uterine segment could carry an increased risk of PPH. At the same time, an unscheduled cesarean delivery should result in significant blood loss, being a surgical intervention with a character of emergency. Surprisingly, our results demonstrate that earlier gestational age was a protective factor for PPH in twins. This is probably due to the fact that preterm delivery is commonly associated with a lower neonatal weight, which, in turn, determines that the uterus is not overstretched and has a greater contractile capacity compared to the one at term.

Other variables normally influencing the risk of PPH, such as the presence of a previous caesarean delivery, advanced maternal age, chorionicity, or maternal and gestational comorbidities, have not shown any significant association with PPH in our population.

To the best of our knowledge, the role of maternal and peripartum factors in the increased risk of PPH in multiple pregnancy is still unclear. What is certainly known is that this risk is significantly increased in multiple pregnancies compared to singletons due to a mechanical factor like uterine overdistension, which can compromise myometrial contractility after delivery. Our study demonstrates that all the physiological changes related to maternal adaptation to multiple gestation could play a pivotal role in increasing the risk of PPH in twins.

These considerations can be fundamental in counseling patients with twin pregnancies in order to discuss the harms and the risks of both vaginal and cesarean delivery not only from the fetal side but also from the maternal side.

The strengths of our study are, above all, the large sample and the selection of the included population, the wide spectrum of gestational ages taken into consideration, and the lack of similar studies in the literature about twin pregnancies. The major limitation is, of course, its retrospective nature. Another limitation is that, unfortunately, in cesarean delivery with intact amniotic membranes, the amount of amniotic fluid summed to the blood loss is higher, and the EBL may not be as accurate as in the vaginal one [21].

The correct identification of factors associated with PPH in twins could ideally allow the possibility of modifying the clinical management in order to positively affect the rate of complications. We still believe that the obstetric management of twin pregnancies after 32 weeks of gestation should be guided by several factors, such as maternal parity, pregnancy characteristics (chorionicity, comorbities, fetal presentation, mother’s desire, the estimated fetal weight, or intertwin weight discordance) and that vaginal birth should be offered if the first twin is vertex, and no other complications are seen. However, we highlight that in the cases of at term pregnancies with appropriate for gestational age or large for gestational age babies, it is recommended to counsel patients regarding her risk of blood loss in labor and to implement strategies aiming at reducing the risk of PPH. It is also very important to understand that EBL should not be considered a crucial factor in deciding the mode of delivery, as our data show that CS does not carry an increased risk of PPH per se, as seen for singletons.

However, more studies are needed to demonstrate whether the use of specific treatment protocols after the risk stratification can lead to an improvement in perinatal outcomes in such pregnancies.

## 5. Conclusions

The results of our study show that the performance of episiotomy—in association with the operative vaginal delivery—and the neonatal weight are the factors that most impact the risk of PPH in twin pregnancies. The latter can be explained with the overdistension of the uterus, which can represent a mechanical cause of PPH.

## Figures and Tables

**Figure 1 diagnostics-13-00446-f001:**
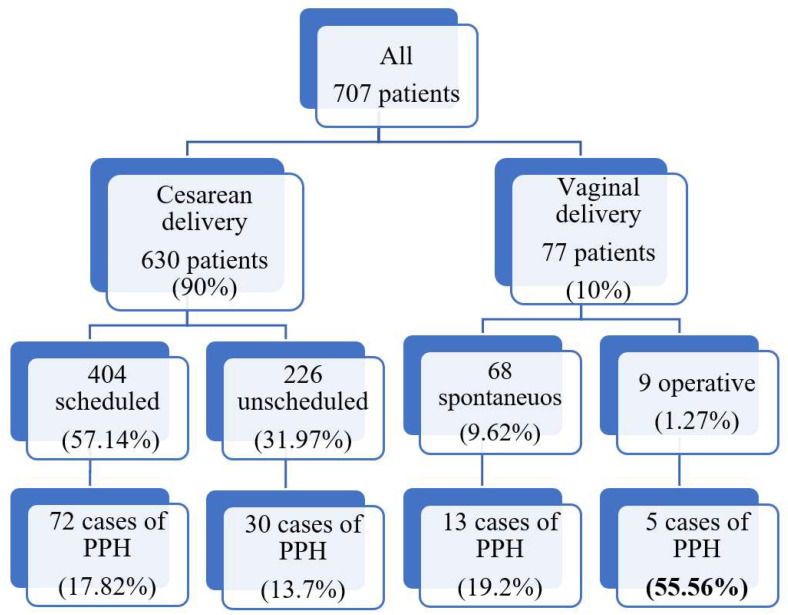
Association between PPH and mode of delivery.

**Table 1 diagnostics-13-00446-t001:** Maternal and peripartum characteristics of all women and separately for subjects with and without postpartum hemorrhage (PPH). *p*-values obtained by chi-square or Fisher exact test for categorical variables and *t*-test or Wilcoxon test for numerical ones are reported. The odds ratios (ORs) obtained by univariate logistic regression model using the presence of HPP as outcome are also shown with their 95% confidence intervals [95% CI].

	All (*n* = 707)	No PPH (*n* = 587)	Yes PPH (*n* = 120)	*p*-Value	OR [95% CI]
**Race**					
White	655 (92.91)	543 (92.82)	112 (93.33)	0.3752	1
Black	24 (3.40)	22 (3.76)	2 (1.67)		0.59 [0.14–2.60]
Other	26 (3.69)	20 (3.42)	6 (5.00)		1.89 [0.72–4.91]
**Age (years)**					
≤35	390 (55.24)	327 (55.80)	63 (52.50)	0.7272	1
35–40	191 (27.05)	158 (26.96)	33 (27.50)		1.26 [0.77–2.04]
>40	125 (17.71)	101 (17.24)	24 (20.00)		1.18 [0.67–2.10]
Mean (SD)	35.17 (5.96)	35.10 (5.94)	35.53 (6.03)	0.4635	
**Parity**					
0	468 (66.20)	382 (65.08)	86 (71.67)	0.2763	1
1	189 (26.73)	164 (27.9)	25 (20.83)		0.64 [0.38–1.09]
>1	50 (7.07)	41 (6.98)	9 (7.50)		1.09 [0.49–2.45]
**Previous vaginal delivery**					
Yes	160 (22.63)	135 (23.00)	25 (20.83)	0.6056	1.07 [0.64–1.78]
**Previous cesarean delivery**					
Yes	86 (12.16)	76 (12.95)	10 (8.33)	0.1589	0.39 [0.16–0.91]
**Smoking (*n* = 664)**					
Yes	104 (15.66)	83 (14.95)	21 (19.27)	0.2576	1.53 [0.89–2.61]
**Comorbidity**					
Hypertension	9 (1.27)	5 (0.85)	4 (2.167)		
Diabetes mellitus	6 (0.85)	4 (0.68)	2 (1.67)		
Pulmunary problems	6 (0.85)	5 (0.85)	1 (0.83)		
Renal/liver problems	3 (0.42)	1 (0.17)	2 (1.67)		
**Pregnancy comorbidity**					
Gestational hypertension	70 (9.93)	56 (9.57)	14 (11.67)	0.4847	1.55 [0.82–2.92]
Gestational diabetes	115 (16.29)	89 (15.19)	26 (21.67)	0.0799	1.59 [0.95–2.68]
**Preclampsia**					
Yes	41 (5.82)	34 (5.81)	7 (5.83)	0.9927	1.20 [0.52–2.81]
**N° of SGA newborn**					
0	427 (60.40)	354 (60.31)	73 (60.83)	0.4721	1
1	187 (26.45)	152 (25.89))	35 (29.17)		1.10 [0.68–1.78]
2	93 (13.15)	81 (13.80)	12 (10.00)		0.79 [0.40–1.58]
**Delivery prior to 36 + 6 weeks**					
Yes	449 (63.51)	384 (65.42)	65 (54.17)	**0.0197**	0.62 [0.40–0.95]
**Perineal tear**					
Episiotomy (before first twin)	30 (4.29)	18 (3.09)	12 (10.26)	**0.0021**	4.08 [1.76–9.46]
Spontaneous vaginal tear	31 (4.43)	27 (4.64)	4 (3.42)		1.09 [0.36–3.24]
No	638 (91.27)	537 (92.27)	101 (86.32)		1
**Type of delivery**					
Cesarean scheduled	404 (57.14)	332 (56.56)	72 (60.00)	**0.007**	0.92 [0.48–1.77]
Cesarean unscheduled	226 (31.97)	196 (33.39)	30 (25.00)		0.65 [0.32–1.33]
Operative vaginal (for both twins)	9 (1.27)	4 (0.68)	5 (4.17)		**5.28 [1.24–22.46]**
Vaginal (ref)	68 (9.62)	55 (9.37)	13 (10.83)		
**Need for blood transfusion**					
Yes	15 (2.13)	3 (0.51)	12 (10.17)	**<0.0001**	
**Medium neonatal weight**					
Medium (SD) grams	2280.06 (391.08)	2261.4 (384.62)	2371.02 (410.70)	**0.0051**	1.001 [1.000–1.001]

**Table 2 diagnostics-13-00446-t002:** Variables associated with PPH after univariate analysis expressed as numbers and rate.

	All (*n* = 707)	PPH (*n* = 120)
**Categories of delivery**		
Scheduled cesarean delivery	404 (57.14%)	72 (17.82%)
Unscheduled cesarean delivery	226 (31.97%)	30 (13.7%)
Spontaneous vaginal delivery	68 (9.62%)	13 (19.2%)
Operative vaginal delivery	9 (1.27%)	5 (55.56%)

**Table 3 diagnostics-13-00446-t003:** Logistic multivariable model using the presence of postpartum hemorrhage (PPH) as outcome. Odds ratios (OR) and 95% confidence intervals are reported both with the *p*-value of models.

	OR [95% CI]	*p*-Value
**Perineal tear**		
Episiotomy	3.59 [1.67–7.74]	0.0037
Spontaneous vaginal tear	0.73 [0.25–2.15]	
No	1	
**Mean neonatal weight**	1.001 [1–1.001]	0.0051

## Data Availability

The data presented in this study are available on request from the corresponding author. The data are not publicly available due to privacy.

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
