# Peer review of "Multiple Pregnancy and the Risk of Postpartum Hemorrhage: Retrospective Analysis in a Tertiary Level Center of Care"

_diagnostics, 2023, doi:10.3390/diagnostics13030446_

Round 1
Reviewer 1 Report
1. Overall interesting cohort
2. Introduction could be shortened with some info moved to discussion; terms are used that should be corrected cesarean delivery is the correct term remove all c section terms
3. M/M could be better organized and presented with primary outcome brought forward as one point.
4. Results: Figure 1 should be before Table 1 and 2; Table 1 category should be Delivery Prior to 36+6 weeks; the episiotomy / operative delivery needs to be better explained as timing of episiotomy before twin 1 or twin 2 ; was all operative delivery for twin 2 as not clearly stated; you use both spon. vaginal tear, perineal tear, and laceration be consistent in both tables; was the vaginal tear associated with operative VD or with spontaneous delivery; why was BMI not in comorbidity section as associated with PPH; indicate in Table 1 with OVD twin 1 or 2 or both
5. Discussion needs to be better organized as the reader gets lost in the process. Need to create a counselling point sheet with the information:
such as spon PTB decreases risk of PPH
PPH risk with each type of delivery Scheduled CD 18%
Unscheduled CD 14%
SVD 19%
PPH Risk with OVD highest at 56% and give the reasons
Other morbidities associated with OVD first twin or second or both
(vaginal -cervical tear; episiotomy; etc)
Discussion second twin management
Reviewer 2 Report
I found the manuscript particularly interesting, but many issues should be fixed before it is ready for publication. Here are my observations.
A) Please, starting from the abstract, avoid using the generic “major blood loss”; instead, always use PPH and be consistent throughout the manuscript. In addition also, in the abstract, please specify that PPH is intended as blood loss >1000ml. This point is important because there are different cut-offs in the literature, and major or severe can range from 1000 to >1500 or >2000 ml.
B) The methods should be better explained by how you collected the amount of blood loss.
C) The methods should also better explain how you handled the fetal weights. I imagine that the population size was the number of pregnancies, and in the logistic regression model, you used the mean weight of the two newborns. This issue should be better explained in the methods. And it should be extended beyond the sensitivity analysis of the logistic regression model (Also, the data in the manuscript and other tables seem to be handled this way). In addition, why did you choose the mean (or max) of the two weights and not the sum (that better represents overdistention)? The methods should also include a more detailed explanation of why you chose to use the mean.
D) In Table 1, in the last two rows, the terms medium and mean are confusing concerning the same measure. Please amend.
E) References 10 and 11 are duplicated.
F) I do not believe that your data support this statement: “It is also very important to state that EBL is not a crucial factor in deciding the mode of delivery as our data show that CS does not carry an increased risk of PPH per se, as seen for singletons.” The cut-off for post-partum hemorrhage is not uncommonly differently considered in vaginal deliveries and cesarean deliveries (CD) because usually, in CD with intact amniotic membranes, the amount of amniotic fluid summed to the blood loss is higher [1]. In your series, apparently, you did not divide the blood loss from amniotic fluid. This topic should be at least discussed in the limits of the study. Eventually can be interesting to show the red blood cell count, hematocrit, and hemoglobin data before and after delivery.
G) I suggest limiting the conclusions to the evidence that answered your aims. Meanwhile, the possible clinical implications can be discussed under other discussion sections [2]: the meaning of the study (including implications for clinicians or policymakers) or unanswered questions and future research.
H) English language should be deeply revised by a native speaker proficient in medical writing.
References
1) Lagrew D, McNulty J, Sakowski C, Cape V, McCormick E, Morton CH. Improving Health
Care Response to Obstetric Hemorrhage, a California Maternal Quality Care Collaborative
Toolkit, 2022.
2) Skelton JR, Edwards SJ. The function of the discussion section in academic medical writing. BMJ. 2000 May 6;320(7244):1269–70.
Round 2
Reviewer 1 Report
Thankyou as most concerns were corrected and the manuscript is improved greatly.
Reviewer 2 Report
The manuscript is significantly improuved and suitable to be published.